# Causal relationships between sex hormone traits, lifestyle factors, and osteoporosis in men: A Mendelian randomization study

Hui Wang[1,2], Jianwen Cheng[1], Donglei Wei[1], Hong Wu[3], Jinmin Zhao[1,2]*

1 Department of Orthopaedic Trauma and Hand Surgery, The First Affiliated Hospital of Guangxi Medical University, Nanning, Guangxi, China, 2 Research Center for Regenerative Medicine, Guangxi Key Laboratory of Regenerative Medicine, Guangxi Medical University, Nanning, Guangxi, China, 3 Department of Medical Research, Affiliated Tumor Hospital of Guangxi Medical University, Nanning, Guangxi, China

* csgkswk@126.com

**Data Availability Statement:** All relevant data are within the paper and its Supporting information files.

## Abstract

Although observational studies have explored factors that may be associated with osteoporosis, it is not clear whether they are causal. Osteoporosis in men is often underestimated. This study aimed to identify the causal risk factors associated with bone mineral density (BMD) in men. Single nucleotide polymorphisms (SNPs) associated with the exposures at the genome-wide significance (p < 5x10$^{-8}$) level were obtained from corresponding genome-wide association studies (GWASs) and were utilized as instrumental variables. Summary-level statistical data for BMD were obtained from two large-scale UK Biobank GWASs. A Mendelian randomization (MR) analysis was performed to identify causal risk factors for BMD. Regarding the BMD of the heel bone, the odds of BMD increased per 1-SD increase of free testosterone (FT) (OR = 1.13, P = 9.4 × 10$^{-17}$), together with estradiol (E2) (OR = 2.51, P = 2.3 × 10$^{-4}$). The odds of BMD also increased with the lowering of sex-hormone binding globulin (SHBG) (OR = 0.87, P = 7.4 × 10$^{-8}$) and total testosterone (TT) (OR = 0.96, P = 3.2 × 10$^{-2}$) levels. Regarding the BMD of the lumbar spine, the odds of BMD increased per 1-SD increase in FT (OR = 1.18, P = 4.0 × 10$^{-3}$). Regarding the BMD of the forearm bone, the odds of BMD increased with lowering SHBG (OR = 0.75, P = 3.0 × 10$^{-3}$) and TT (OR = 0.85, P = 3.0 × 10$^{-3}$) levels. Our MR study corroborated certain causal relationships and provided genetic evidence among sex hormone traits, lifestyle factors and BMD. Furthermore, it is a novel insight that TT was defined as a disadvantage for osteoporosis in male European populations.

## Introduction

Osteoporosis is a kind of bone disease that is associated with aging. It is characterized by reduced bone mass and damage to the bone microstructure, resulting in increased bone fragility and easy fracture. With the aging of society, the prevalence of osteoporosis is increasing significantly, which places a heavy burden on patients, their families and society. In general, the diagnosis and treatment of osteoporosis is more focused on women than men. Indeed, women

**Funding:** The author(s) received no specific funding for this work.

**Competing interests:** The authors have declared that no competing interests exist.

over the age of 50 are four times more likely to develop osteoporosis and two times more likely to develop osteopenia than men [1]. The most common and serious complication of osteoporosis is fracture, mainly in the distal forearm, proximal humerus, thoracic and lumbar vertebrae, and proximal femur. The incidence of distal forearm, hip and spinal fractures is lower in males than in females [2]. However, studies have shown that men have more osteoporosis-related complications and higher mortality after osteoporosis fractures than women [3, 4]. The first-year mortality from hip fracture was 38% in men versus 28% in women [5]. Giuseppe Rinonapoli's review of the literature found that osteoporosis in men is poorly documented, underreported, and underdiagnosed. This underestimation can have serious consequences and high life risks [6].

According to some relevant literature reports [6–8], there are many possible risk factors for osteoporosis in men, including alcoholism, body mass index (BMI), glucocorticoid overdose, hypogonadism, parathyroidism, hyperthyroidism, gastro-intestinal diseases, and hypercalciuria. However, the causal relationships between these risk factors and osteoporosis have not been fully established. Mendelian randomization (MR) is an emerging epidemiological causal inference method that has achieved great success in determining risk factors for diseases. MR is used to evaluate the potential causal influences of risk factors on outcomes by using genetic instrumental variables (IVs) and can reduce the bias caused by confounders or reverse causation [9]. A recent study used MR to identify a number of potential causal risk factors for osteoporosis, including fasting insulin levels, type 2 diabetes, fasting glucose levels, hip and waist circumference adjusted for BMI, and HDL cholesterol levels [10]. Additionally, another MR study suggested a null association between depression and osteoporosis [11]. However, the causal relationship between sex hormones and osteoporosis has not been reported. Here, we included seven major risk factors, including sex hormones and lifestyle factors, to explore their causal relationships with male osteoporosis. The ultimate goal of this MR was to elucidate the causal relationships between sex hormones and male osteoporosis.

## Materials and methods

### Risk factors from a genome-wide association study (GWAS)

The seven major risk factors were divided into two categories: sex hormone traits and lifestyle factors. We extracted the instrumental variables (IVs) for sex hormone traits from a recent genome-wide association study (GWAS) [12]. This was a large-scale meta-analysis conducted using the UK Biobank. This study included 425,097 individuals of European ancestry with sex-hormone binding globulin (SHBG), total testosterone (TT), and estradiol (E2) data and 382,988 with free testosterone (FT) data. The statistics of this study were adjusted for age, partially adjusted for BMI, and disaggregated by sex. In our MR analysis, we included only SHBG, TT, FT, and E2 levels in men as the instrumental variables of hormone traits.

People may have many lifestyle factors, and smoking, drinking and diet may be considered the most common factors. Given the amount of food eaten should be related to body size. BMI was included in this study as a diet-related factor. So the lifestyle factors included men's smoking status, drinking status and BMI. The GWAS summary statistics for smoking and drinking included 1,232,091 individuals of European ancestry for smoking and 941,280 for drinking [13]. In this study, smoking status was divided into four types, and the research content of drinking was the number of drinks per week. Smoking initiation and the number of drinks per week were selected as the IVs for smoking and drinking, respectively, in our MR analysis. The GWAS summary statistics of the men's BMIs were from the Genetic Investigation of Anthropometric Traits (GIANT) consortium, which included 339,224 European individuals and adjusted for age, age squared, sex and four genotype-based principal components.

### GWAS statistics of BMD

Osteoporosis is characterized by reduced bone density and an increased risk of fracture. Bone mineral density (BMD), which represents bone strength, is often used in the clinical diagnosis of osteoporosis [14, 15]. Therefore, BMD statistics were used to represent osteoporosis in our MR analysis. The GWAS summary statistics of BMD were from two large meta-analyses that reported genetic variants [16, 17]. One study included 53236 European individuals for lumbar spine BMD (LS-BMD), femoral neck BMD (FN-BMD), and forearm BMD (FA-BMD) and adjusted for sex, age and weight. The other GWAS included 426824 European individuals for heel BMD (HE-BMD) and adjusted for age, sex and genotype. To avoid bias caused by overlap of exposure and outcome datasets, the GWAS statistics of BMD as outcome did not include participants from the UK Biobank.

### Mendelian randomization design and statistical analysis

Mendelian randomization should be performed under three basic assumptions: (1) genetic variations are strongly associated with exposure; (2) genetic variations are not associated with either known or unknown confounders; and (3) genetic variations are independent of the outcome except by means of exposure. We included SNPs that achieved genome-wide significance (p value $< 5 \times 10^{-8}$) and a minor allele frequency $> 0.01$ in the GWAS datasets for each IV risk factor. The linkage disequilibrium (LD) of the significant SNPs was set to meet $r2 = 0.01$ and $KB = 5000$. The phenotypic variation explained by SNPs was calculated as follows: $R2 = 2 \times beta2 \times (1-EAF) \times EAF/SD2$, with EAF = effect allele frequency and beta = the effect of each SNP on the exposures [18]. The F statistic ($F = beta2/se2$) was used to test the strength of the association between these SNPs and the exposure factors. SNPs with strong statistical power (F statistics$>10$) were included. Fig 1 shows the MR design framework. S1 Data shows the characteristics of the SNPs selected to be significantly associated with exposures.

Prior to the MR analysis of the two samples, we unified the effect value directions of the exposure data and outcome data and removed the SNPs that were palindromic with the intermediate allele frequencies [19]. We used inverse-variance weighted (IVW) method as the main estimation method for the MR statistical analysis to examine the causal relationships between

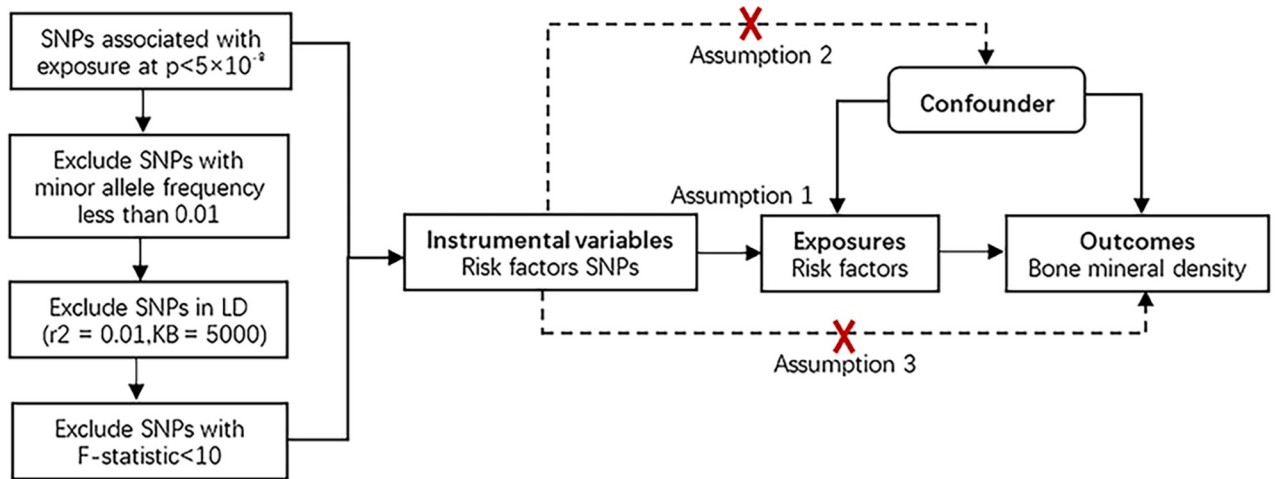

**Fig 1. The framework of our Mendelian randomization study.** Assumption 1: The genetic variations are strongly associated with exposure; Assumption 2: The genetic variations are not associated with either known or unknown confounders; Assumption 3: SNPs should influence risk of the outcome through the exposure, not through other pathways.

the exposure factors and BMD. In addition, weighted-median [20], MR–Egger [21] and MR-PRESSO [22] methods were used as supplements and for sensitivity analyses. The weighted-median method provides consistent estimation results as long as the weight of the valid instrumental variables is greater than or equal to 50% [20]. We used Cochrane's Q value to assess the heterogeneity and the MR-Egger intercept [21] to detect horizontal pleiotropy. Leave-one-out analysis was utilized to estimate the possibility of the results being driven by a single SNP. When outliers were detected by the MR-PRESSO method, they were removed, and the MR causal estimation was recalculated. When the MR-PRESSO corrected results existed, they were taken as the main MR-PRESSO results. When heterogeneity existed, the weighted median was adopted as the main effect method. The false discovery rate (FDR) based on the Benjamini and Hochberg method was used to adjust the P values for multiple testing. The mRnd (https://cnsgenomics.shinyapps.io/mRnd/) was used to calculate the statistical power of MR.

All Mendelian randomization analyses were performed in R software version 4.1.1 using the "TwoSampleMR" [19], "MR-PRESSO" [22], and "MendelianRandomization" [23] packages.

## Results

The number of SNPs that were closely related to the exposure factors, ranging from 13 to 319, after LD with other variants or that were absent from the LD reference panel were removed. Their explained variances varied from 0.04% to 6.9%. The F statistics for each SNP and the general F statistics were all greater than 10 (Table 1).

For HE-BMD, univariable Mendelian randomization analysis suggested that higher FT, E2, and BMI levels could increase HE-BMD and were considered protective factors for osteoporosis. However, higher SHBG and TT levels might reduce HE-BMD and were recognized as risk factors for osteoporosis (P<0.05, after FDR control) (Fig 2A, S1 Table in S1 File). The odds of HE-BMD increased per 1-SD increase in FT (OR = 1.13, P = $9.4 \times 10^{-17}$), E2 (OR = 2.51, P = $2.3 \times 10^{-4}$), and BMI (OR = 1.06, P = $3.6 \times 10^{-2}$) levels. In addition, a 1-SD increase in SHBG could help reduce HE-BMD (OR = 0.87, P = $7.4 \times 10^{-8}$), together with TT (OR = 0.96, P = $3.2 \times 10^{-2}$). Smoking and drinking were not associated with an increase in the odds of HE-BMD (P>0.05, Fig 2A). The outliers detected by the MR-PRESSO method were removed, and the MR causal estimation was recalculated. There was heterogeneity in the SHBG level, TT level, FT level, smoking status, drinking status and BMI. No horizontal pleiotropy was found for any of the risk factors. The leave-one-out method for the SHBG, TT, FT, E2, and BMI levels indicated that no instrumental variables influenced the causal inference, and their

**Table 1. Summary of risk factors.**

| Exposure | NSNP | Sample | $R^2$(%) | F | people | PMID |
|----------|------|--------|----------|------|--------|------|
| SHBG | 250 | 425,097 | 2.3 | 40.0 | European,Male | 32042192 |
| TT | 157 | 425,097 | 6.9 | 200.6 | European,Male | 32042192 |
| FT | 80 | 382,988 | 3.3 | 163.3 | European,Male | 32042192 |
| E2 | 13 | 425,097 | 0.04 | 13.4 | European,Male | 32042192 |
| Smoking | 319 | 1,232,091 | 4.7 | 190.4 | European, Male, Female | 30643251 |
| Drinking | 84 | 941,280 | 0.6 | 67.6 | European, Male, Female | 30643251 |
| BMI | 34 | 339,224 | 1.7 | 172.5 | European,Male | 25673413 |

NSNP number of single nucleotide polymorphism, $R^2$ phenotype variance explained by genetics, F F statistics, PMID ID of publication in PubMed, SHBG sex hormone binding globulin, TT total testosterone, FT free testosterone, E2 estradiol, BMI body mass index.

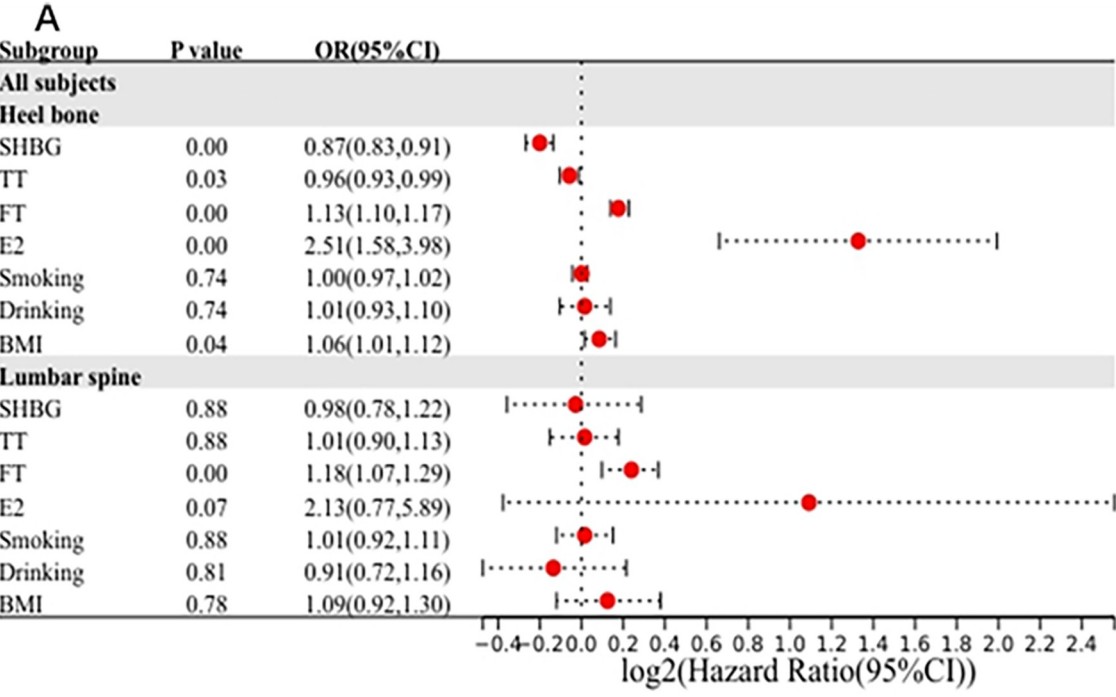

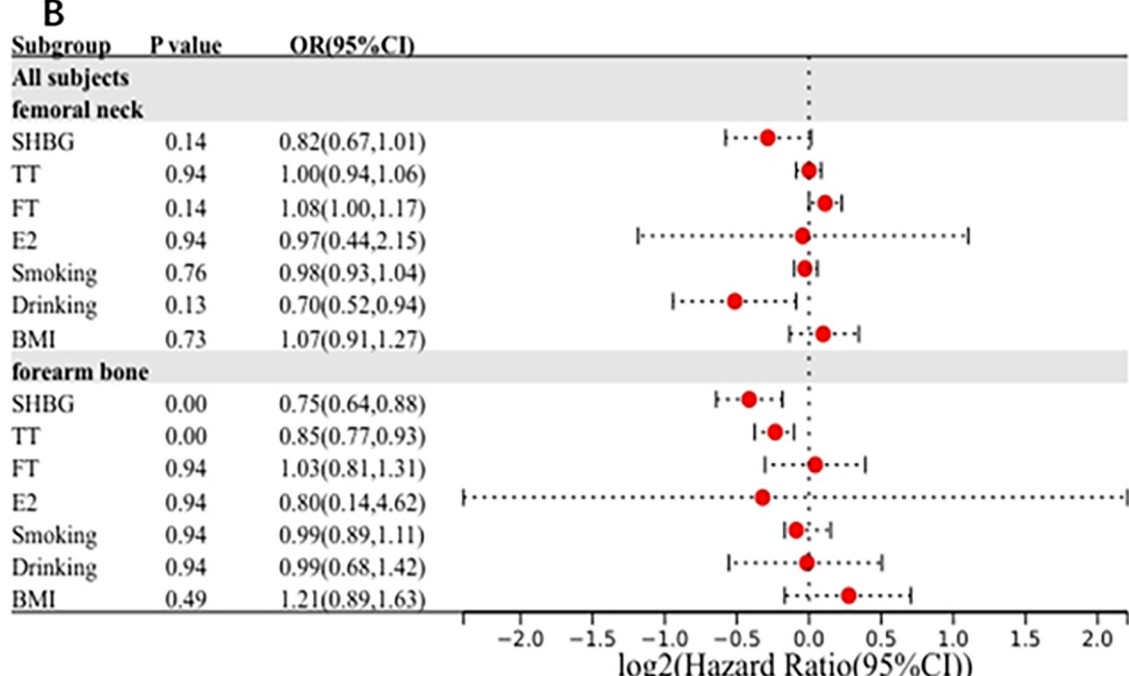

**Fig 2. The forest plot of Mendelian randomization results.** A is the results from bone mineral density of heel bone and lumbar spine; B is the results from bone mineral density of femoral neck and forearm bone.

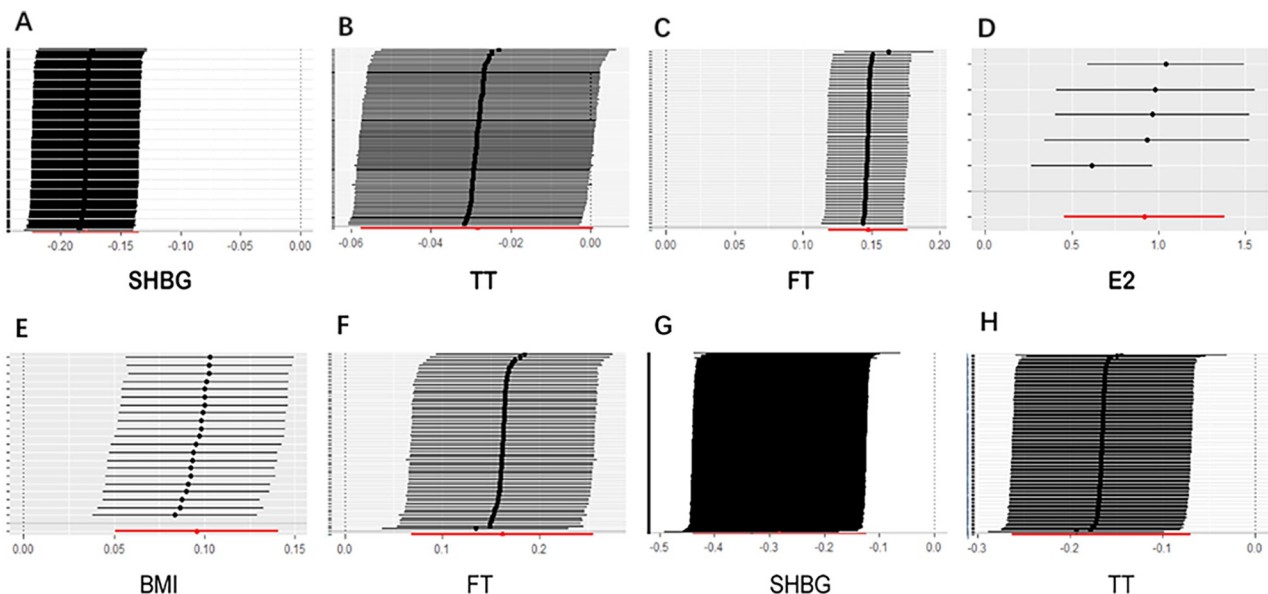

**Fig 3. The leave-one-out method to verify the robustness of Mendelian randomization results.** A-E are the leave-one-out results of SHBG, TT, FT, E2 and BMI from bone mineral density of heel bone respectively. F is the leave-one-out result of FT from bone mineral density of lumbar spine; G and H are the leave-one-out results of SHBG, TT from bone mineral density of forearm bone.

MR results were quite robust (Fig 3). For the existing heterogeneity, the statistics from the weighted-median approach were consistent with those of the IVW models, except for the TT model.

For LS-BMD, univariable Mendelian randomization analysis suggested that higher FT levels could increase LS-BMD and was considered a protective factor for osteoporosis (P<0.05, after FDR control) (Fig 2A, S1 Table in S1 File). The odds of LS-BMD increased per 1-SD increase in FT (OR = 1.18, P = $4.0 \times 10^{-3}$). SHBG levels, TT levels, E2 levels, smoking status, drinking status and BMI levels were not associated with an increase in the odds of LS-BMD (P>0.05, Fig 2A). There was heterogeneity in SHBG levels, TT levels and smoking status. No horizontal pleiotropy was found for any of the risk factors. The leave-one-out method for FT levels indicated that no instrumental variables influenced the causal inference, and its MR results were quite robust (Fig 3). For the existing heterogeneity, the statistics from the weighted-median approach were consistent with those of the IVW models except for the SHBG model. The weighted-median approach result for SHBG levels was not associated with LS-BMD (OR = 0.98, P = 0.876).

For FN-BMD, all factors, including the SHBG level, the TT level, the FT level, the E2 level, smoking status, drinking status and BMI level, were not associated with an increase in the odds of FN-BMD (P>0.05, Fig 2B). There was heterogeneity in the SHBG level and drinking status. No horizontal pleiotropy was found for any of the risk factors. For FA-BMD, higher SHBG and TT levels might reduce FA-BMD and were recognized as risk factors for osteoporosis (P<0.05, after FDR control) (Fig 2B, S1 Table in S1 File). A 1-SD increase in SHBG could help reduce FA-BMD (OR = 0.75, P = $3.0 \times 10^{-3}$), together with TT (OR = 0.85, P = $3.0 \times 10^{-3}$). SHBG levels, TT levels, FT levels, E2 levels, smoking status, drinking status and BMI levels were not associated with increased odds of FA-BMD (P>0.05, Fig 2B). Only the FT level showed heterogeneity. No horizontal pleiotropy was found for any of the risk factors. The leave-one-out method for SHBG and TT levels indicated that no instrumental variables influenced the causal inference and that their MR results were quite robust (Fig 3).

**Table 2. Causal relationships of factors on HE-BMD by multivariable MR.**

| Exposure | Outcome | NSNPs | P |
|---|---|---|---|
| SHBG | HE-BMD | 106 | 0.015 |
| TT | HE-BMD | 83 | 0.025 |
| FT | HE-BMD | 38 | 0.055 |
| BMI | HE-BMD | 18 | 0.028 |

NSNP number of single nucleotide polymorphism, P p-value of OR, SHBG sex hormone binding globulin, TT total testosterone, FT free testosterone, BMI body mass index.

Multiple influencing factors were found in HE-BMD through univariable MR analysis, we conducted multivariable MR to verify the independence of the effects identified. Since the number of estrogen-related SNPs was small, the number of SNPs was 0 after multivariate exclusion, estrogen was not included in multivariate MR analysis. We performed multivariate analysis of SHBG, TT, FT, BMI and HE-BMD, and after correction FT was no longer significantly causally related. SHBG, TT and BMI are considered to be independent influencing factors of HE-BMD(P<0.05, Table 2).

The original results of IVW, weighted-median, MR–Egger and MR-PRESSO results for HE-BMD, LS-BMD, FN-BMD and FA-BMD can be found in S2 Table in S1 File, together with the heterogeneity and pleiotropy tests. The statistical power for HE, LS, and FA outcomes were all greater than 90%, except that the power of BMI in HE outcomes was 60%.

## Discussion

Our MR study explored the causal relationships between sex hormone traits, lifestyle factors and HE, LS, FN, and FA bone mineral density in men. We identified SHBG, TT, FT, E2 and BMI levels as potential causal factors for HE-BMD; FT for LS-BMD; and SHBG and TT for FA-BMD. In addition, SHBG, TT and BMI were considered to be independent influencing factors of HE-BMD. This study provides insights into the fact that higher SHBG and TT levels can decrease the BMD of HE and FA; higher FT levels can increase the BMD of HE and LS; and higher E2 and BMI levels can increase HE-BMD. These results imply that a) the higher the SHBG and TT levels are, the more severe the outcome of osteoporosis; b) higher FT, E2 and BMI levels may protect against osteoporosis in men. Smoking and alcohol consumption were not observed to be associated with osteoporosis in men.

Sex hormone binding globulin (SHBG), a plasma glycoprotein, binds with high affinity to sex steroids (5α-dihydrotestosterone, testosterone, and 17β-estradiol) to regulate their bioavailability and access to target cells. SHBG levels have been linked to a number of diseases, including osteoporosis [24]. The association between SHBG and bone mineral density has been investigated in several cross-sectional studies in different countries [25–27]. Two studies in Chinese and US populations showed an increased risk of osteoporosis with increased SHBG levels. Zha et al. investigated the correlation between SHBG and testosterone levels in blood and the BMD of hip bones and LS in Chinese men over 45 years of age and found that the SHBG level was negatively correlated with BMD [25]. In the US population, a cross-sectional study of 6,434 participants aged 18–80 years from the National Health and Nutrition Examination Survey (NHANES) from 2013 to 2016 found that SHBG levels were significantly associated with LS-BMD and that SHBG levels increased the predictive value of bone loss in adults [26]. In Koreans, SHBG levels were negatively correlated with the BMD of the pelvis but not with the BMD of other regions [27]. However, our study found that SHBG levels had a causal

relationship with HE and FA, and that SHBG levels were negatively correlated with BMD, while no causal relationship was found for LS and FN. The results of a recent Mendelian randomization study between SHBG levels and BMD, were consistent with ours [28].

Another causal and negative association with BMD in our MR study was TT levels. Evidence linking TT levels and BMD in adults is limited. There are controversies about whether TT is correlated with bone mineral density. In a previously mentioned Korean study, TT was reported to be positively correlated with bone mineral density in the ribs, LS, and FA [27]. Li et al. found that TT levels were not associated with vertebral trabecular BMD in middle-aged and elderly Chinese men [29]. A cross-sectional study involving a noninstitutionalized U.S. population sample from the National Health and Nutrition Examination Survey found that the correlation between TT levels and total BMD varied by sex and race. The correlation between total testosterone levels and total bone mineral density in female adolescents was not significant, which was also the case in males, adults aged 40 to 60 years, and other racial/ethnic groups. In non-Hispanic blacks, total testosterone was inversely associated with total bone mineral density at concentrations greater than 500 ng/dl [30]. In our MR study involving European men, a causal relationship was found between TT levels and the bone mineral density of HE and FA, and there was a negative correlation between them. This result of our MR study implies that elevated TT levels may induce decreased bone mineral density and even lead to osteoporosis, which is also a novel insight from other studies.

Several studies have shown that FT and E2 are beneficial for bone mineral density in both men and women [25, 31, 32]. One of the studies included 102 male patients with chronic kidney disease who generally have lower sex hormone levels. The results indicated that FT levels were positively correlated with the BMD of the LS, hips and FN, and E2 levels were positively correlated with the BMD of the LS and FN [31]. Our study confirmed a causal relationship between FT levels, E2 levels and BMD in men through Mendelian randomization. FT was positively correlated with the BMD of the HE and LS, and was not an independent factor in HE-BMD; E2 was positively correlated with HE-BMD.

The relationship between BMI, obesity and bone mineral density investigated in many studies did not show a linear correlation but the correlation varied with BMI. A large British cross-sectional study found a positive association between obesity and BMD in normal-weight men but found a negative association in men with elevated BMI levels. In women, a negative association between premenopausal obesity and BMD was observed, but a positive association was observed in postmenopausal women [33]. Another study of the US population showed that the relationship between BMI and BMD was not simply linear and that a saturation value existed. The saturation effect analysis showed that the BMI saturation value was 26.13 (kg/m2) for femur bone and 26.82 (kg/m2) for LS. The results suggest that maintaining a BMI at a slightly overweight level (approximately 26 kg/m2) may obtain optimal BMD [34]. A study on the relationship between the obesity index and BMD in Chinese people found that BMI levels were positively correlated with LS-BMD in men aged over 60 years [35]. In our MR study, we found a positive causal relationship between BMI levels and HE bone mineral density in men. However, its statistical power was low (60%), and there may be bias in predicting the true causal relationship. In the above studies, BMI was found to be positively correlated with BMD within a certain range but was found to be negatively correlated with BMD beyond this range. Our study results are consistent with those of two other MR analyses of the correlation between BMI and BMD [36, 37].

The effects of smoking and alcohol consumption on BMD have been controversial in previous studies. Some observational studies suggest that smoking and drinking may reduce bone density [38–40]. Elisa et al. found that current smokers had significantly greater BMD volume loss in trabeculae than those who never smoked [38]. The number of pack-years of smoking

was found to be negatively correlated with total hip BMD in middle-aged Korean men [39]. Patients with alcohol dependence have a significantly lower BMD and higher incidence of osteoporosis than healthy people [40]. However, some studies have found no correlation between smoking, alcohol consumption and bone mineral density [41–43]. The third National Health and Nutrition Examination Survey of noninpatients in the United States found that femoral neck bone mineral density was numerically lower in smokers than in never-smokers, but this was not statistically significant after controlling for confounding factors [41]. Another study exploring risk factors for bone mineral density in US residents found that there was no consensus on the impact of smoking and alcohol consumption on BMD [42]. In this study, we explored the causal relationship between smoking status, drinking status and BMD from the perspective of genetic predispositions. Similar to the conclusion of another MR analysis [44], we did not find a causal relationship between smoking, alcohol consumption and BMD.

## Conclusions

In conclusion, by MR approach analysis, we identified SHBG and TT levels as potential causal risk factors for BMD loss in men that may increase the incidence rate of osteoporosis. It is a novel insight that TT levels differ among ethnic groups and are defined as a disadvantage in male European populations. FT and E2 levels are considered to be potentially beneficial causal factors for BMD in men and to help prevent osteoporosis. Although a causal relationship has been found between BMI and BMD, it remains to be explored due to its low statistical power.

Our study was an MR design study, which uses genotypes as instrumental variables to infer the association between phenotypes and diseases and is suitable for causal inference. All participants in the study were of European ancestry, so our results were not biased by population stratification. However, certain factors may affect statistical power—an insufficiently large numbers of cases of BMD and different characteristics of the included population. For example, several studies observed a nonlinear relationship between BMI and BMD. BMI was positively correlated with BMD within a limit range but was negatively correlated with BMD above this limit (excessive obesity). Although our study found a positive causal relationship between BMI and BMD, the result may be affected by these factors. Second, observational studies found different correlations between TT levels and BMD in different ethnic groups. In this study, TT had a negative causal relationship with BMD, which could only be applied to European populations. In addition to this result, the rest of the results should be extended to other populations with caution because the participants included in our study were all European.

## Supporting information

**S1 File.**
(DOCX)

**S1 Data. The characteristics of the SNPs selected to be significantly associated with exposures.**
(XLSX)

## Acknowledgments

The Genetic Factors for Osteoporosis Consortium and Genetic Investigation of Sex Hormone Traits and Lifestyle Factors Consortium were obtained from large-scale GWAS summary -level data. The authors thank all investigators for sharing these data.

## Author Contributions

**Conceptualization:** Hui Wang.

**Data curation:** Hui Wang.

**Formal analysis:** Hui Wang.

**Methodology:** Jianwen Cheng.

**Project administration:** Jianwen Cheng.

**Resources:** Donglei Wei, Hong Wu.

**Software:** Donglei Wei.

**Supervision:** Jinmin Zhao.

**Validation:** Hong Wu.

**Writing – original draft:** Hui Wang.

**Writing – review & editing:** Hui Wang, Jinmin Zhao.

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
