## [Decision Letter · Decision Letter 0]

6 Jun 2022

PONE-D-22-13525Causal relationships between sex hormone traits, lifestyle factors, and osteoporosis in men: A Mendelian randomization studyPLOS ONE

Dear Dr. zhao,

Thank you for submitting your manuscript to PLOS ONE. After careful consideration, we feel that it has merit but does not fully meet PLOS ONE’s publication criteria as it currently stands. Therefore, we invite you to submit a revised version of the manuscript that addresses the points raised during the review process.

 To be considered for publication, the authors should fully address the comments received.

We look forward to receiving your revised manuscript.

Kind regards,

Renato Polimanti, Ph.D.

Academic Editor

PLOS ONE

Journal Requirements:

Additional Editor Comments:

Wang and colleagues conducted an interesting Mendelian randomization analysis to test the effect linking sex hormones, lifestyle factors, and osteoporosis in men. Although the analyses are generally adequate. There are major issues that the authors should address:

1. The authors should clarify whether there is sample overlap among the datasets investigated because this could introduce a bias in the analyses performed.

2. The authors performed a less stringent clumping than that usually applied in the MR field (see PMID: 31448343). They should clarify why they made this decision and whether this may have confounded the results of the analyses conducted.

3. The authors reported GSCAN sample sizes as including 23andMe data. 23andMe data are not publicly available. The authors should clarify whether they obtained these data. If 23andMe were not obtained, they should report the correct sample size.

4. It would be important to perform a multivariable MR to verify the independence of the effects identified.

Reviewers' comments:

Reviewer's Responses to Questions

**Comments to the Author**

1. Is the manuscript technically sound, and do the data support the conclusions?

Reviewer #1: Yes

2. Has the statistical analysis been performed appropriately and rigorously? 

Reviewer #1: Yes

3. Have the authors made all data underlying the findings in their manuscript fully available?

Reviewer #1: No

4. Is the manuscript presented in an intelligible fashion and written in standard English?

Reviewer #1: Yes

5. Review Comments to the Author

Reviewer #1: Causal relationships between sex hormone traits, lifestyle factors, and osteoporosis in men: A Mendelian randomization study

This study assessed the potential causal relationship of osteoporosis in men with sex hormones and lifestyle-related factors. The subject of this study is interesting and important. Based on the methods described, I am satisfied that the study was appropriately designed and conducted. I have only minor comments for author’s consideration in improving the presentation of this study.

First, it is not clear what informed the choice of the lifestyle factors assessed in this study. This is important given there are many similar factors that have been associated with the outcome of interest. How did author trim down the ‘lifestyle’ factors to arrive at the one assessed here, or was it just a random selection? Further, I would not call BMI a lifestyle factor.

Second, lines 65 – 66: ‘However, the true causal relationships between these risk factors and osteoporosis have not been fully established.’ The use of ‘true’ here may send a different message, especially as MR cannot be said to be able to assess ‘true’ causal effect. The question that can arise would be has there been any assessment of the causal relationship between these factors and osteoporosis (in men in particular). If yes, state it and highlight the gap the present study fills. If not, then I suggest authors should remove ‘true’ as this can be controversial.

Third, lines 106 – 107: ‘Therefore, BMD statistics were used to represent osteoporosis in our MR analysis.’ To what extent does the BMD represent osteoporosis? Any reference to support this?

Fourth, the presentation/description of the results can be difficult to follow, understandably because of the many factors, and MR models. I see the Tables and Figures are much clearer, especially, Table S1. Can authors follow a similar approach in describing their results and provide a sub-section for each of the exposure variables (as in Table S1). This will enhance the clarity of this work and enable readers to comprehend the findings. The same applies to the discussion section, authors can use a similar approach without the need for sub-sections.

6. PLOS authors have the option to publish the peer review history of their article (what does this mean?). If published, this will include your full peer review and any attached files.

Reviewer #1: **Yes: **Dr Emmanuel Adewuyi

---

## [Author Response · Author response to Decision Letter 0]

20 Jun 2022

Replies to the reviewers’ comments:

Reviewer #1: Additional Editor Comments

1. The authors should clarify whether there is sample overlap among the datasets investigated because this could introduce a bias in the analyses performed.

Response：The analysis data for exposure in this study were from UK Biobank and published papers, and the outcome data were from published papers, excluding UK Biobank , so the overlap between exposure and outcome was avoided. We added an explanation in the article, on lines 114-116.

2. The authors performed a less stringent clumping than that usually applied in the MR field (see PMID: 31448343). They should clarify why they made this decision and whether this may have confounded the results of the analyses conducted.

Response：An important step in MR studies is to remove SNPS with linkage disequilibrium(LD). The LD is mainly using two parameters, r2 and KB. R2 is the data between 0 and 1. R2 =1 indicates that there is a complete LD between two SNPS. R2 =0 indicates that there is a complete linkage equilibrium between two SNPS. KB is the length of the region of linkage imbalance. With the decrease of R2 and the increase of KB, more and more SNPs with LD will be removed, and less and less IV will be left. Generally, the less the number of IV is, the less promiscuous and pluripotency exists, but the corresponding statistical power may be insufficient. The most common default parameters in literature are (r2=0.001, kb=10000) and (r2=0.01, kb=5000). Parameter (r2=0.01, kb=5000) is set moderately, and this parameter is also used in some published articles, such as PMID: 35267896 and PMID: 35140703. It is believed that this parameter setting will not confound the analysis results.

3. The authors reported GSCAN sample sizes as including 23andMe data. 23andMe data are not publicly available. The authors should clarify whether they obtained these data. If 23andMe were not obtained, they should report the correct sample size.

Response：IVs associated with smoking and alcohol status were identified in this study from the GSCAN Consortium. The data obtained by us comes from the article PMID: 30643251 (https://doi.org/10.1038/s41588-018-0307-5), in which sample size is introduced and relevant SNPs information can be obtained in the attached table. In addition, we have attached a new excel table (file S1) for all SNPs characteristics as Ivs in exposure. Line130-131.

4. It would be important to perform a multivariable MR to verify the independence of the effects identified.

Response：In this study, different factors were used to explore the causal relationship between BMD at four sites. Multiple factors were found only in the HE-BMD as the outcome, so the multivariable analysis in this section has been added in accordance with the editor's comments (lines 225-237). Some changes have also been made to lines 249-250 and 300 in the discussion section.

Reviewer #2: Review Comments

1. First, it is not clear what informed the choice of the lifestyle factors assessed in this study. This is important given there are many similar factors that have been associated with the outcome of interest. How did author trim down the ‘lifestyle’ factors to arrive at the one assessed here, or was it just a random selection? Further, I would not call BMI a lifestyle factor.

Response：Indeed, there may be many lifestyle factors. In this paper, we want to explore the three most common factors: smoking, drinking and diet. In view of the fact that the body shape is the most relevant to the amount of food consumed, BMI was included into the lifestyle factors in this study. The corresponding explanatory text is added to lines 92-94 of the article.

2. Second, lines 65 – 66: ‘However, the true causal relationships between these risk factors and osteoporosis have not been fully established.’ The use of ‘true’ here may send a different message, especially as MR cannot be said to be able to assess ‘true’ causal effect. The question that can arise would be has there been any assessment of the causal relationship between these factors and osteoporosis (in men in particular). If yes, state it and highlight the gap the present study fills. If not, then I suggest authors should remove ‘true’ as this can be controversial.

Response: The reviewer's suggestion was very accurate, and we have deleted the "true" (line 65) according to the reviewer's suggestion.

3. Third, lines 106 – 107: ‘Therefore, BMD statistics were used to represent osteoporosis in our MR analysis.’ To what extent does the BMD represent osteoporosis? Any reference to support this?

Response: Osteoporosis is characterized by reduced bone density and an increased risk of fracture. Bone mineral density (BMD), which represents bone strength, is often used in the clinical diagnosis of osteoporosis[14,15]. In several Mendelian randomized analyses of osteoporosis, BMD was also used instead of osteoporosis. References are: PMID: 34105796; PMID: 35646051; PMID: 33439309; PMID: 34259888.

4. Fourth, the presentation/description of the results can be difficult to follow, understandably because of the many factors, and MR models. I see the Tables and Figures are much clearer, especially, Table S1. Can authors follow a similar approach in describing their results and provide a sub-section for each of the exposure variables (as in Table S1). This will enhance the clarity of this work and enable readers to comprehend the findings. The same applies to the discussion section, authors can use a similar approach without the need for sub-sections.

Response: Table S1 has summarized the results of 7 exposure factors corresponding to BMD at 4 different sites used as outcome. For each of the exposure variables, a new excel table (file S1) is attached to show the characteristics of each SNP because there are a large number of SNPs selected as being closely related to 7 exposures. Line 130-131

---

## [Decision Letter · Decision Letter 1]

11 Jul 2022

Causal relationships between sex hormone traits, lifestyle factors, and osteoporosis in men: A Mendelian randomization study

PONE-D-22-13525R1

Dear Dr. zhao,

We’re pleased to inform you that your manuscript has been judged scientifically suitable for publication and will be formally accepted for publication once it meets all outstanding technical requirements.

Kind regards,

Renato Polimanti, Ph.D.

Academic Editor

PLOS ONE

Additional Editor Comments (optional):

The authors adequately addressed the main comments made by reviewers and no further changes are needed.

Reviewers' comments:

Reviewer's Responses to Questions

**Comments to the Author**

1. If the authors have adequately addressed your comments raised in a previous round of review and you feel that this manuscript is now acceptable for publication, you may indicate that here to bypass the “Comments to the Author” section, enter your conflict of interest statement in the “Confidential to Editor” section, and submit your "Accept" recommendation.

Reviewer #1: (No Response)

2. Is the manuscript technically sound, and do the data support the conclusions?

Reviewer #1: Yes

3. Has the statistical analysis been performed appropriately and rigorously? 

Reviewer #1: Yes

4. Have the authors made all data underlying the findings in their manuscript fully available?

Reviewer #1: Yes

5. Is the manuscript presented in an intelligible fashion and written in standard English?

Reviewer #1: Yes

6. Review Comments to the Author

Reviewer #1: Author has largely addressed all my comments.

It appears, though, from authors' response, that my comment no 4 did not come across clearly. What I requested was for author to describe their results (in the results section) according to the main headings, possibly using subsections. This will enhance clarity of the study. However, if this is not possible, I think, I am satisfied overall. The Tables and Figures are helpful.

Just a few minor comments as follows:

1. While I note that Table 1 clarified whether the GWAS was for male or female, this detail needs to be clearly stated in the methods section as well. Can author state this information for each of the GWAS in the relevant section of the methods.

2. Also, author need to state the limitation(s) of including female GWASs (for Smoking and drinking) since this study is focused on men (male).

3. Lastly, authors need to read over their conclusion both in the abstract and the main manuscript. I expect a concise take home message from this study, especially, in line with the argument of the authors regarding the importance of the study.

7. PLOS authors have the option to publish the peer review history of their article (what does this mean?). If published, this will include your full peer review and any attached files.

Reviewer #1: **Yes: **Dr Emmanuel Adewuyi

---

## [Editor Report · Acceptance letter]

15 Jul 2022

PONE-D-22-13525R1 

Causal relationships between sex hormone traits, lifestyle factors, and osteoporosis in men: A Mendelian randomization study 

Dear Dr. Zhao:

I'm pleased to inform you that your manuscript has been deemed suitable for publication in PLOS ONE. Congratulations! Your manuscript is now with our production department. 

Kind regards, 

on behalf of

Dr. Renato Polimanti 

Academic Editor

PLOS ONE